# Prevalence and risk factors of birth asphyxia at Livingstone University teaching hospital

Nestorine N. Ngongo[1,2], Patson Sichamba[1], Natasha Chishala[1,2],
Mulenga D. Chibeka[1,2], Mighty Chimba[1,2], Simon Kacha[1,2], Kakula Simutowe[1,2],
Prince Mulambo[1,2], Emmanuel O. Riwo[1,2], Salma M. Baines[1,2],
Kimberley R. Kurehwatira[1,2], Hanzooma Hatwiko[1,2], Chileleko Siakabanze[1,2],
Emmanuel L. Luwaya[1,2], Katongo H. Mutengo[1,2], Lukundo Siame[1,2], Bislom C. Mweene[1,2],
Martin Chakulya[1,2], Joreen P. Povia[3], Sepiso K. Masenga[1,2*]

1 Department of Physiological Sciences, Mulungushi University, School of Medicine and Health Sciences, Livingstone, Zambia, 2 Department of Pathology, Livingstone Center for Prevention and Translational Science, Livingstone, Zambia, 3 Department of Health Economics, Livingstone Center for Prevention and Translational Science, Livingstone, Zambia

* sepisomasenga@lcpts.org; sepisomasenga@gmail.com

## Abstract

Birth asphyxia remains a leading cause of neonatal mortality in low-resource settings, with Zambia reporting a rate of 24 deaths per 1,000 live births. Birth asphyxia accounts for over 20% of neonatal ICU admissions. This study aimed to determine its prevalence and risk factors to inform targeted interventions. We conducted a secondary data analysis of medical records at Livingstone University Teaching Hospital, Zambia, including 497 maternal–neonatal records of deliveries between 15 July 2024 and 31 March 2025. Data were abstracted from the Obstetrics and Gynaecology Department and Neonatal Intensive Care Unit between 1 and 20 April 2025 using REDCap. The primary outcome was birth asphyxia, defined as failure to establish spontaneous respiration with Apgar ≤5 at 5 minutes or hypoxic-ischemic encephalopathy. Maternal, obstetric, and neonatal demographic and clinical variables were collected. Bivariate and multivariable logistic regression were used to identify factors associated with birth asphyxia, with statistical significance set at $p < 0.05$. The prevalence of birth asphyxia was 6.8% (34/497). Significant correlates included eclampsia (adjusted odds ratio [AOR]=17.3; 95% CI:2.7-111.0; $p = 0.002$), foetal distress (AOR = 7.3; 95% CI:2.5-20.9; $p < 0.001$), and resuscitation with suction (AOR = 3.8; 95% CI:1.2–11.5; $p = 0.018$) or facial oxygen (AOR = 3.5; 95% CI:1.0-11.6; $p = 0.044$). Neonates requiring bag-mask ventilation had 65.6% asphyxia rates versus 5% without ($p < 0.001$). Post-term gestation (15.2% asphyxia) and abnormal foetal heart rates (28.6%) were also associated with higher risk. The 6.8% asphyxia prevalence at LUTH reflects regional disparities, with eclampsia and foetal distress being critical modifiable risks. Strengthening emergency obstetric care, foetal monitoring, and neonatal resuscitation capacity could reduce preventable cases. These findings

**Data availability statement:** All relevant data are within the paper and its Supporting Information files.

**Funding:** The authors received no specific funding for this work.

**Competing interests:** The authors have declared that no competing interests exist.

underscore the need for context-specific strategies to improve perinatal outcomes in Zambia and similar settings.

## Introduction

Birth asphyxia, defined as the failure to initiate and sustain breathing at birth, remains a leading cause of neonatal mortality and long-term neurodevelopmental impairment worldwide [1]. Globally, an estimated 24% of neonatal deaths are attributed to birth asphyxia, with the burden disproportionately affecting low- and middle-income countries (LMICs) [2]. In sub-Saharan Africa, birth asphyxia accounts for nearly 30% of neonatal mortality, reflecting critical gaps in perinatal care [3]. Zambia mirrors this troubling trend, with birth asphyxia representing the second leading cause of neonatal death after prematurity, contributing significantly to the country's neonatal mortality rate of 24 deaths per 1,000 live births [4,5]. The high prevalence of birth asphyxia in resource-limited settings like Zambia stems from multiple interrelated factors. Maternal complications such as hypertensive disorders, antepartum haemorrhage, and prolonged labour are well-established risk factors [6,7]. Neonatal characteristics including prematurity, low birth weight, and intrauterine growth restriction further compound the risk [8]. Importantly, health system challenges - including delays in emergency obstetric care, inadequate foetal monitoring, and shortages of skilled birth attendants - exacerbate the problem in LMICs [6,7]. Despite global initiatives like the Every Newborn Action Plan, progress in reducing birth asphyxia has been slow in resource-constrained environments [4]. Recent facility-based surveillance in Zambia reports an incidence of 8.5 cases per 1,000 live births at tertiary centres, significantly higher than the national average of 6.2 per 1,000 [8]. This disparity underscores the need for context-specific research to identify modifiable risk factors and optimize care delivery. Current evidence suggests that up to 60% of birth asphyxia cases in similar settings may be preventable through timely interventions [9].

Risk factors for birth asphyxia are multifactorial and often interrelated. Maternal determinants, such as hypertensive disorders, prolonged labour, and antepartum haemorrhage, are well-documented contributors [9,10]. Neonatal factors, including prematurity, low birth weight, and intrauterine growth restriction, further compound the risk. Health system challenges, such as delays in accessing emergency obstetric care, inadequate foetal monitoring, and shortages of skilled birth attendants, exacerbate the problem in LMICs [6,11]. In Zambia, socioeconomic disparities, limited prenatal care coverage, and cultural barriers to facility-based deliveries further heighten vulnerability to adverse birth outcomes [12].

Epidemiologically, the incidence of birth asphyxia at LUTH aligns with regional patterns. A 2022 hospital-based surveillance report noted an incidence rate of 8.5 cases per 1,000 live births, higher than the national average of 6.2 per 1,000 [1]. This disparity underscores the need for context-specific analyses to identify modifiable risk factors and optimize care delivery. While global frameworks, such as the Every Newborn Action Plan, emphasize reducing preventable neonatal deaths, localized data remain critical for tailoring interventions.

Understanding the prevalence and risk factors of birth asphyxia at LUTH is essential for informing clinical protocols, resource allocation, and community health initiatives. This study aims to bridge existing knowledge gaps by providing evidence-based insights into the epidemiology of birth asphyxia in this setting, ultimately contributing to improved neonatal survival and health system resilience in Southern Province and beyond.

## Methodology

### Study design

We employed a secondary data analysis of medical records, utilizing retrospective data extracted from medical records of neonates and mothers admitted to Livingstone University Teaching Hospital (LUTH). The study focused on neonates born between 15th July 2024–31st March 2025. We started the data abstraction on 1st April 2025 and completed by 20th April 2025.

### Study setting

The research was conducted at LUTH, a tertiary referral hospital in Southern Province, Zambia, serving a high-risk obstetric and neonatal population. Data were sourced from the Obstetrics and Gynaecology department and the Paediatric Neonatal Intensive Care Unit (NICU), which manage the majority of perinatal and neonatal emergencies in the region.

### Eligibility and recruitment

A total of 497 maternal-neonatal medical records were systematically reviewed. Inclusion criteria comprised all women who delivered at LUTH during the study period, irrespective of gestational age or delivery mode. Neonates with incomplete medical records or those with major congenital anomalies unrelated to birth asphyxia were excluded to minimize confounding.

### Data collection

A retrospective review of medical records was conducted between 1st April 2025, and 20th April 2025, by trained research assistants using a standardized protocol. The Research Electronic Data Capture (REDCap) platform facilitated structured data abstraction, ensuring accuracy and consistency. Variables collected included maternal demographics, obstetric history, intrapartum events, neonatal resuscitation techniques, and neonatal outcomes.

### Operational definitions

The primary outcome variable, *birth asphyxia*, was operationalized as the failure to establish spontaneous respiration at birth, accompanied by clinical evidence of hypoxic-ischemic encephalopathy (HIE) or an Apgar score ≤5 at 5 minutes, consistent with the definition by Gillam-Krakauer and Gowen Jr (2024) [13]. Independent variables encompassed maternal age, hypertensive disorders, anaemia (haemoglobin <11 g/dL), prematurity (<37 weeks gestation). Fetal distress was defined according to the Royal College of Obstetricians and Gynaecologists (RCOG) Green-top Guidelines as clinical signs indicating significant physiological compromise of the fetus, such as hypoxia and acidosis [14]. This was based on a combination of factors, including a non-reassuring cardiotocograph (CTG). Abnormal cardiotocography CTG was defined per the FIGO 2015 guidelines, encompassing features such as a baseline fetal heart rate <100 or >180 beats per minute, reduced variability (<5 bpm), repetitive late or variable decelerations, or a sinusoidal pattern. Prolonged labour was defined as labour lasting more than 20 hours in primigravida women and more than 14 hours in multiparous women. Prolonged labour was defined as a first stage exceeding 20 hours in primigravidas or 14 hours in multiparous women, and a second stage exceeding 3 hours in primigravidas or 2 hours in multiparous women. The diagnosis of fetal distress was made clinically based on a combination of an abnormal CTG tracing and supportive findings such as significant meconium-stained liquor [14] or fetal scalp blood sampling results indicative of acidaemia [15].

Global Public Health
PLOS

### Data analysis

We exported data from the REDCap application to Microsoft Excel 2013, where it underwent cleaning and coding. Stat-Crunch was then used for data analysis. Descriptive statistics were employed to summarize categorical variables through frequencies and percentages, while continuous variables were summarized using the median and interquartile range. The Shapiro-Wilk test was applied to evaluate data normality. The association between two categorical variables was assessed using the chi-square test, and differences between two medians were evaluated with the Wilcoxon rank-sum test. Bivariate and multivariable logistic regression analyses were conducted to identify factors associated with birth asphyxia. Variables with a p-value < 0.2 in the bivariate analysis were considered for inclusion in the initial multivariable model. A stepwise backward elimination approach was used, and variables with a p-value < 0.05 were retained in the final model. Statistical significance was set at p < 0.05.

### Ethical considerations

Ethical approval was granted by the Mulungushi University School of Medicine and Health Sciences Research Ethics Committee (SMHS-MU2-2024-148) on 09th June, 2024 and LUTH administration gave permission to access patient's records. All data collected and analysed were de-identified to ensure complete confidentiality. No information leading to identification of patients during and after analysis was abstracted and entered in the data collection form. Secondary data was used in this project. A written/verbal consent was not applicable and was therefore waived by the ethics committee. We used the Strengthening the Reporting of Observational Studies in Epidemiology (STROBE) checklist to guide reporting, Supplementary file S1 File.

## Results

From a total of 1216 files available for abstraction, 603 medical records were reviewed, 497 records met the inclusion criteria and were included in the analysis, **Fig 1**. maternal-neonatal medical records were systematically reviewed. The remaining 106 files were excluded due to incomplete information such as missing outcome variable and Neonates with incomplete records or congenital anomalies unrelated to asphyxia.

### Characteristics of the study population

Among 497 maternal-neonatal medical records analysed at LUTH, birth asphyxia occurred in 6.8% (n = 34) of neonates. Mothers had a median age of 26 years (IQR: 20–31), with mothers of infants with asphyxia slightly younger (median 24 years vs. 26 years), **Table 1**. Maternal hypertension was documented in 8.2% (n = 41) of cases, while anaemia affected 17.6% (n = 83), antepartum haemorrhage 5.5% (n = 27), preeclampsia 7% (n = 34), and diabetes mellitus 0.2% (n = 1). Eclampsia was rare (1.8%, n = 9) but present in 33.3% (n = 3) of asphyxia cases. Maternal fever occurred in 3.2% (n = 16), with 25% (n = 4) of febrile mothers delivering infants with asphyxia. Placenta previa was noted in 1.7% (n = 8), with 25% (n = 2) of these cases involving asphyxia. Breech presentation occurred in 5.7% (n = 27), and caesarean deliveries accounted for 13.8% (n = 66) of births. Prolonged labour was documented in 7.8% (n = 38) of cases, with 13.2% (n = 5) of these resulting in asphyxia. Meconium-stained amniotic fluid was observed in 15.4% (n = 76) of deliveries, with 11.8% (n = 9) of these linked to asphyxia. Abnormal foetal heart rate/rhythms were recorded in 7.1% (n = 35) of cases, of which 28.6% (n = 10) involved asphyxia. Prematurity affected 11.9% (n = 58) of neonates, while post-term gestation occurred in 9.3% (n = 46), with 15.2% (n = 7) of post-term infants experiencing asphyxia. Resuscitation interventions included suction in 27.8% (n = 137) of neonates, facial oxygen in 5.3% (n = 26), and bag-mask ventilation in 6.5% (n = 32). Among resuscitated infants, asphyxia rates were 19.0% (n = 26) for suction, 38.5% (n = 10) for facial oxygen, and 65.6% (n = 21) for bag-mask ventilation. Foetal distress was documented in 9.8% (n = 47) of cases, with 38.3% (n = 18) of these infants developing asphyxia.

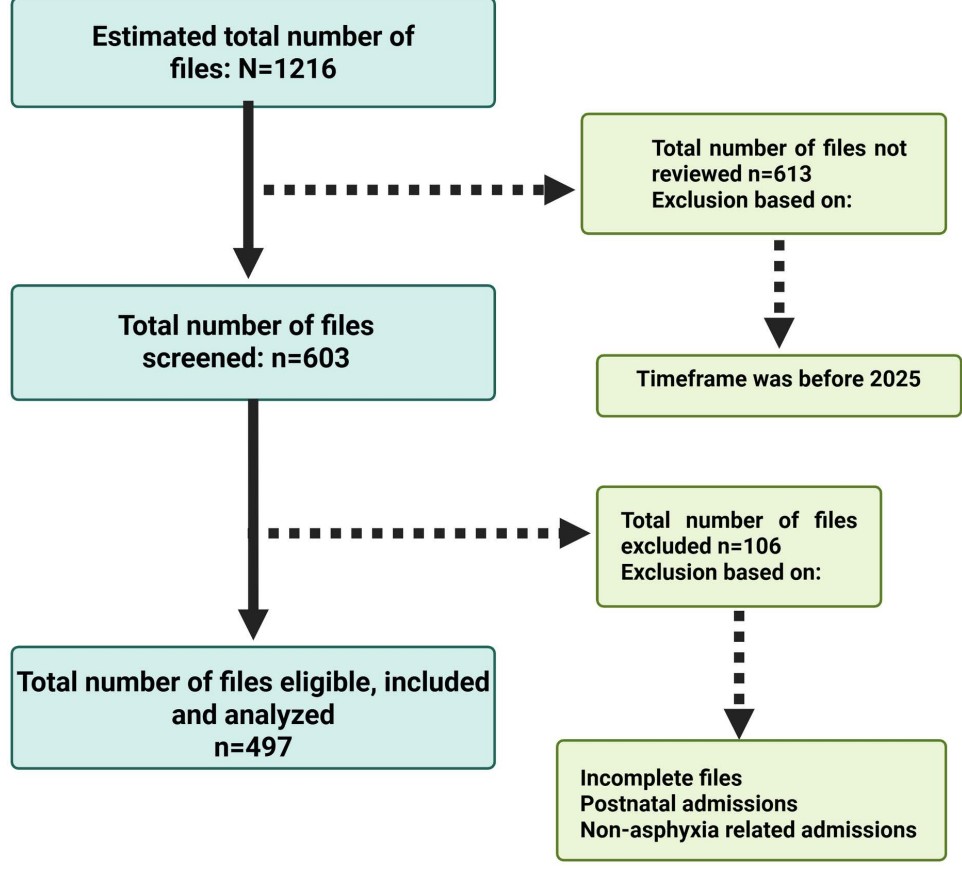

**Fig 1. Eligibility flow diagram.**

### Factors associated with birth asphyxia in logistic regression

Table 2 shows a logistic regression of factors associated with birth asphyxia. In bivariate analyses, four factors emerged as strong predictors of birth asphyxia including eclampsia, foetal distress, and neonatal resuscitation methods. After adjusting for confounders in the final multivariable model, four variables remained independently and significantly associated with birth asphyxia: eclampsia conferred a 17-fold increase in odds (AOR = 17.3; 95% CI: 2.7–111.0), foetal distress a seven-fold increase (AOR = 7.3; 95% CI: 2.5–20.9), suction resuscitation a 3.8-fold increase (AOR = 3.8; 95% CI: 1.2–11.5), and facial-oxygen resuscitation a 3.5-fold increase (AOR = 3.5; 95% CI: 1.0–11.6).

### Discussion

The prevalence of birth asphyxia in this study was 6.8% (34/497), which falls within the range reported by other Zambian tertiary hospitals (6.5-7.1%) but remains higher than the national average of 6.2 per 1,000 live births [16]. This elevated prevalence at a referral facility like Livingstone University Teaching Hospital likely reflects its role in managing high-risk pregnancies and complicated deliveries from surrounding regions. When compared globally, our findings show a 2–4 times higher prevalence than reported in high-income countries (1.5-3%) [17], but are consistent with rates from similar low-resource settings across sub-Saharan Africa where studies report 5–9% prevalence [1]. The relatively stable prevalence observed across multiple Zambian teaching hospitals suggests systemic challenges in perinatal care that warrant

**Table 1. General characteristics of the participants (N = 497).**

| Variable | Median (IQR) or frequency (%) | Yes = 34 (6.8%) | No = 463 (93.2%) | P. value |
|---|---|---|---|---|
| **Mother age, year** *n = 497* | 26(20-31) | 24(19,29) | 26(21,32) | 0.257 |
| **Maternal hypertension** *n = 494* | | | | 0.512 |
| yes | 41 (8.2) | 4 (9.8) | 37 (90.2) | |
| No | 453 (91.7) | 30 (6.6) | 423 (93.4) | |
| **Anaemia** *n = 471* | | | | 0.238 |
| Yes | 83 (17.6) | 3(3.6%) | 80 (96.4) | |
| No | 388 (82.4) | 30 (7.7) | 358 (92.3) | |
| **Antepartum haemorrhage** *n = 487* | | | | 1.000 |
| Yes | 27 (5.5) | 1 (3.7) | 26 (96.3) | |
| No | 460 (94.5) | 32 (7) | 428 (93) | |
| **Preeclampsia** n = 490 | | | | 0.715 |
| Yes | 34 (7) | 1 (2.9) | 33 (97.1) | |
| No | 456 (93) | 31 (6.8) | 425 (93.2) | |
| **Eclampsia** n = 489 | | | | **0.017** |
| Yes | 9 (1.8) | 3 (33.3) | 6 (66.7) | |
| No | 480 (98. 2) | 30 (6.3) | 450 (93.7) | |
| **Diabetes mellitus** *n = 488* | | | | 1.000 |
| Yes | 1 (0.2) | 0 (0) | 1(100) | |
| No | 487 (99.8) | 33 (6.8) | 454 (93.2) | |
| **Placenta previa** | | | | 0.085 |
| Yes | 8 (1.7) | 2(25) | 6(75) | |
| No | 467 (98.3) | 28 (6) | 439 (94) | |
| **Gravidity** *n = 497* | 2(1,4) | 1(1,4) | 2(1,4) | 0.263 |
| **Parity** *n = 496* | 2(1,3) | 1(1,4) | 2(1,3) | 0.276 |
| **History of abortion** *n = 464* | | | | 0.396 |
| Yes | 24 (5.2) | 0 (0) | 24 (100) | |
| No | 440 (94.8) | 32 (7.3) | 408 (92.7) | |
| **Presentation of foetus** *n = 475* | | | | 1.000 |
| *Cephalic* | 448 (94.3) | 32 (7.1) | 416 (92.9) | |
| *Breech* | 27 (5.7) | 1 (3.7) | 26 (96.3) | |
| **Mode of delivery** *n = 478* | | | | 0.170 |
| *Normal vaginal delivery* | 412 (86.2) | 25 (6.1) | 387 (93.9) | |
| *Caesarean section* | 66 (13.8) | 7 (10.6) | 59 (89.4) | |
| **Delivery by C/S** *n = 54* | | | | 0.443 |
| *Elective* | 30 (55.6) | 5 (16.7) | 25(83.3) | |
| *Emergency* | 24 (44.4) | 2(8.3) | 22 (91.7) | |
| **History of prolonged labour** *n = 487* | | | | 0.171 |
| Yes | 38 (7.8) | 5 (13.2) | 33 (86.8) | |
| No | 449 (92.2) | 29 (6.5) | 420 (93.5) | |
| **Umbilical cord prolapse** *n = 479* | | | | 1.000 |
| Yes | 3 (0.6) | 0 (0) | 3 (100) | |
| no | 476 (99.4) | 34 (7.1) | 442 (92.9) | |
| **Cephalopelvic disproportion** *n = 479* | | | | 1.000 |
| Yes | 4 (0.8) | 0 (0) | 4 (100) | |
| No | 475 (99.2) | 34 (7.2) | 441 (92.8) | |
| **Maternal hypotension** *n = 473* | | | | 0.488 |
| Yes | 9 (1.9) | 1 (11.1) | 8 (88.9) | |
| No | 469 (98.1) | 33 (7) | 436 (93) | |

*(Continued)*

Table 1. (Continued)

| Variable | Median (IQR) or frequency (%) | Birth asphyxia Yes = 34 (6.8%) | No = 463 (93.2%) | P. value |
|---|---|---|---|---|
| **Mother had fever** *n = 496* | | | | **0.018** |
| Yes | 16 (3.2) | 4 (25) | 12 (75) | |
| No | 480 (96.8) | 30 (6.2) | 450 (93.8) | |
| **Multiple birth** *n = 491* | | | | 0.623 |
| Yes | 18 (3.7) | 0 (0) | 18 (100) | |
| No | 473 (96.3) | 33 (7) | 440(93) | |
| **Polyhydramnios** *n = 483* | | | | 1.000 |
| Yes | 4 (0.8) | 0 (0) | 4 (100) | |
| No | 479 (99.2) | 34 (7.1) | 442(92.9) | |
| Meconium-stained amniotic fluid *n = 493* | | | | 0.064 |
| Yes | 76 (15.4) | 9 (11.8) | 67 (88.2) | |
| No | 417 (84.6) | 25 (6) | 392 (94) | |
| **Cardiac arrhythmia** *n = 492* | | | | **<0.0001** |
| Yes | 35 (7.1) | 10 (28.6) | 25 (71.4) | |
| No | 457 (92.9) | 24 (5.2) | 433(94.8) | |
| **Growth retardation** *n = 490* | | | | 1.000 |
| Yes | 2(0.4) | 0 (0) | 2 (100) | |
| No | 488 (99.6) | 34 (7) | 454(93) | |
| **Premature delivery** *n = 489* | | | | 0.404 |
| Yes | 58 (11.9) | 2 (3.4) | 56 (96.6) | |
| No | 431 (88.1) | 31 (7.2) | 400(92.8) | |
| **Child resuscitated** | | | | |
| *With suction n = 492* | | | | **<0.0001** |
| Yes | 137 (27.8) | 26 (19) | 111(81) | |
| No | 355 (72.2) | 8 (2.3) | 347(97.7) | |
| *With facial oxygen n = 492* | | | | **<0.0001** |
| Yes | 26 (5.3) | 10 (38.5) | 16 (61.5) | |
| No | 466 (94.7) | 24 (5.1) | 442(94.9) | |
| *With Bag + mask n = 491* | | | | **<0.0001** |
| Yes | 32 (6.5) | 21 (65.6) | 11 (34.4) | |
| No | 459(93.5) | 23 (5) | 436(95) | |
| **Nuchal cord** *n = 493* | | | | 0.097 |
| Yes | 17 (3.4) | 3 (17.7) | 14(82.3) | |
| No | 476 (96.6) | 30 (6.3) | 446(93.7) | |
| **Foetal distress** *n = 480* | | | | **<0.0001** |
| Yes | 47 (9.8) | 18 (38.3) | 29 (61.7) | |
| No | 433 (90.2) | 15 (3.5) | 418(96.5) | |
| **Gestational age of baby at birth** *n = 493* | | | | 0.033 |
| *Pre-term* | 102 (20.7) | 8 (7.6) | 94 (92.2) | |
| *Term* | 345 (70) | 18 (5.2) | 327 (94.8) | |
| *Post-term* | 46 (9.3) | 7 (15.2) | 39 (84.8) | |
| **Baby weight**, kg *n = 493* | 3(2.7,3.4) | 3.1(2.71,3.5) | 3(2.7, 3.3) | 0.225 |

Note: The total N for each variable varies due to missing data in the medical records.

Abbreviations: IQR; interquartile range, Kg; kilogram, CS; Caesarean section,

**Table 2. Bivariate and multivariate logistic regression analysis of factors associated with birth asphyxia.**

| | Bivariate | | Multivariate | |
| --- | --- | --- | --- | --- |
| Variables | OR (95%CI) | p. value | AOR (95%CI) | p. value |
| Age of mother | 0.4 (0.9, 1.0) | 0.367 | | |
| Eclampsia | 7.5 (1.8, 31.4) | **0.005** | 17.3 (2.7,111.0) | **0.002** |
| Mother had fever | 5 (1.5, 16.4) | **0.008** | 2.6 (0.5,13.4) | 0.231 |
| Abnormal heart rate/rhythm | 7.2 (3.1,16.7) | **<0.0001** | 1.3 (0.4, 4.5) | 0.618 |
| Child resuscitated with suction | 10.2 (4.5,23.1) | **<0.0001** | 3.8 (1.2,11.5) | **0.018** |
| Child resuscitated with facial oxygen | 11.5 (4.7, 28.0) | **<0.0001** | 3.5 (1.0, 11.6) | **0.044** |
| Child resuscitated with bag+mask | 9.9 (4.3, 23.0) | **<0.0001** | 1.5 (0.5, 4.7) | 0.505 |
| Foetal distress | 17.3 (7.9, 37.8) | **<0.0001** | 7.3 (2.5, 20.9) | **0.0004** |
| Gestational age at birth | 1.4 (0.7, 2.7) | 0.355 | | |

OR; odds ratio, AOR; adjusted odds ratio, CI; confidence interval

further investigation and targeted interventions to reduce preventable birth asphyxia cases [18]. These prevalence patterns highlight the ongoing need to strengthen maternal and neonatal care services in resource-limited settings to achieve global targets for reducing preventable neonatal mortality. Three key findings merit particular attention in the context of existing literature.

In our study, we observed a strongest association between eclampsia with birth asphyxia. This is consistent with established literature linking hypertensive disorders to foetal hypoxia. A study by Traub et al. also found results which aligns with our study findings with eclampsia been one of the causes of foetal hypoxia complications and mortalities [19]. Furthermore, another study Gathiram and Moodley suggest that eclampsia disrupts placental blood flow, leading to acute or chronic oxygen deprivation and death [20]. The 33.3% asphyxia rate among mothers with eclampsia in our cohort compared to 6.3% in cases of mothers without- reinforces calls for improved detection and management of hypertensive disorders in pregnancy. These finding gains particular urgency given that 60% of these cases may be preventable with timely intervention.

Furthermore, the significant association between the need for specific resuscitation measures at birth and birth asphyxia was also noteworthy. We found that there was requirement for active resuscitation interventions immediately after birth is a direct indicator of compromised neonatal condition, including inadequate oxygenation consistent with birth asphyxia. These findings highlight the critical role of timely and effective neonatal resuscitation in managing the consequences of intrapartum events leading to birth asphyxia as indicated by kariuki et al [21]. Additionally, other study finding suggests that Meconium aspiration and respiratory depression were common culprits, which necessitated skilled resuscitation at birth [22]. Another study Idris et al. also mirrors similar.

Another variable that was significantly associated with birth asphyxia was foetal distress, reinforcing its role as a proxy for intrapartum hypoxia. A study by Eller et al. found that abnormal foetal heart rate patterns often precede asphyxia, necessitating immediate obstetric intervention [23]. Additionally, a study by Ekblom et al. also found results which aligns with our findings suggesting that poor quality intrauterine care precipitates foetal distress and finally mortality. Another study Kebede et al. also echoes similar results suggesting that foetal distress is one of the complications that leads to birth asphyxia [24].

## Strengths

This study provides valuable insights into the prevalence and risk factors of birth asphyxia at a major Zambian referral hospital. A key strength is the robust methodological approach, including the use of standardized data collection via the

REDCap platform, which minimized errors and improved data accuracy. The study's relatively large sample size (n = 497) enhances the reliability of the findings, particularly in assessing rare but critical conditions like eclampsia. Additionally, the inclusion of both maternal and neonatal variables allowed for a comprehensive evaluation of risk factors, strengthening the multivariable logistic regression analysis. The use of adjusted odds ratios (AORs) helps account for confounding variables, providing more precise estimates of association. Furthermore, the study's focus on a tertiary hospital setting where high-risk pregnancies are managed offers insights into severe cases of birth asphyxia that may not be captured in community-based studies.

Our findings indicated that a prolonged second stage of labour was not an independent predictor of umbilical cord acidaemia in the final multivariate regression model. This appears to contrast with several contemporary studies that have quantified a time-dependent increase in the risk of neonatal metabolic acidaemia with a prolonged second stage, particularly when complicated by pathological FHR patterns [25–27]. The established physiological rationale is that recurrent, prolonged cord compression and head compression during maternal pushing can lead to progressive fetal hypoxia and metabolic acidosis [25–27].

The discrepancy between our results and this established literature may be explained by the management protocol in our setting. It is plausible that clinicians, upon recognizing a pathological FHR pattern (a strong predictor in our study), proactively intervened with expedited delivery via operative vaginal or cesarean delivery. This timely intervention could interrupt the cascade towards significant fetal acidaemia, thereby weakening the direct statistical association between the duration of the second stage and the ultimate umbilical cord pH in our cohort. Therefore, our findings may reflect the effectiveness of intrapartum surveillance and intervention in mitigating the risks associated with a prolonged second stage, rather than contradicting the underlying pathophysiology.

### Limitations

Despite its strengths, this study has several limitations. The retrospective design introduces potential biases, such as incomplete or inconsistent documentation in medical records, particularly for variables like foetal distress timing and resuscitation details. The single-centre nature of the study limits generalizability to primary healthcare facilities, where most Zambian births occur and resources may be even more constrained. Additionally, wide confidence intervals for some associations (e.g., eclampsia AOR: 2.7-111.0) indicate reduced precision due to the low prevalence of certain exposures, which may affect the interpretation of risk magnitudes. The study also did not account for socioeconomic factors, transportation delays, or care-seeking behaviours, which are known to influence birth outcomes in low-resource settings. Finally, the 9-month study period may not have captured seasonal variations in birth asphyxia incidence.

### Conclusion

This study highlights the persistent challenge of birth asphyxia in Zambia, with a 6.8% prevalence at a tertiary hospital that exceeds national averages and reflects broader patterns in resource-limited settings. The identification of eclampsia, foetal distress, and resuscitation needs as key risk factors underscores critical intervention points for reducing preventable neonatal mortality. While the retrospective design and single centre focus present limitations, these findings provide actionable evidence to strengthen perinatal care through improved management of hypertensive disorders, enhanced foetal monitoring, and expanded neonatal resuscitation capacity. The results call for targeted health system improvements that align with global new-born survival initiatives, emphasizing the urgent need to translate these findings into clinical practice and policy to achieve sustainable reductions in birth asphyxia and its devastating consequences in Zambia and similar settings.

### Supporting information

**S1 File. Strobe checklist.**
(DOCX)

**S1 Data. Data.**
(XLSX)

## Author contributions

**Conceptualization:** Nestorine N. Ngongo, Sepiso K. Masenga.

**Data curation:** Nestorine N. Ngongo, Martin Chakulya, Sepiso K. Masenga.

**Formal analysis:** Martin Chakulya, Sepiso K. Masenga.

**Investigation:** Sepiso K. Masenga.

**Methodology:** Nestorine N. Ngongo, Sepiso K. Masenga.

**Project administration:** Sepiso K. Masenga.

**Software:** Martin Chakulya, Sepiso K. Masenga.

**Supervision:** Patson Sichamba, Martin Chakulya, Sepiso K. Masenga.

**Validation:** Nestorine N. Ngongo, Patson Sichamba, Natasha Chishala, Mulenga D. Chibeka, Mighty Chimba, Simon Kacha, Kakula Simutowe, Prince Mulambo, Emmanuel O. Riwo, Salma M. Baines, Kimberley R. Kurehwatira, Hanzooma Hatwiko, Chileleko Siakabanze, Emmanuel L. Luwaya, Katongo H. Mutengo, Lukundo Siame, Bislom C. Mweene, Martin Chakulya, Joreen P. Povia, Sepiso K. Masenga.

**Visualization:** Nestorine N. Ngongo, Patson Sichamba, Natasha Chishala, Mulenga D. Chibeka, Mighty Chimba, Simon Kacha, Kakula Simutowe, Prince Mulambo, Emmanuel O. Riwo, Salma M. Baines, Kimberley R. Kurehwatira, Hanzooma Hatwiko, Chileleko Siakabanze, Emmanuel L. Luwaya, Katongo H. Mutengo, Lukundo Siame, Bislom C. Mweene, Martin Chakulya, Joreen P. Povia, Sepiso K. Masenga.

**Writing – original draft:** Nestorine N. Ngongo, Patson Sichamba, Natasha Chishala, Mulenga D. Chibeka, Mighty Chimba, Simon Kacha, Kakula Simutowe, Prince Mulambo, Emmanuel O. Riwo, Salma M. Baines, Kimberley R. Kurehwatira, Hanzooma Hatwiko, Chileleko Siakabanze, Emmanuel L. Luwaya, Katongo H. Mutengo, Lukundo Siame, Bislom C. Mweene, Martin Chakulya, Joreen P. Povia, Sepiso K. Masenga.

**Writing – review & editing:** Nestorine N. Ngongo, Patson Sichamba, Natasha Chishala, Mulenga D. Chibeka, Mighty Chimba, Simon Kacha, Kakula Simutowe, Prince Mulambo, Emmanuel O. Riwo, Salma M. Baines, Kimberley R. Kurehwatira, Hanzooma Hatwiko, Chileleko Siakabanze, Emmanuel L. Luwaya, Katongo H. Mutengo, Lukundo Siame, Bislom C. Mweene, Martin Chakulya, Joreen P. Povia, Sepiso K. Masenga.

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
