## [Decision Letter · Decision Letter 0]

29 Aug 2025

PGPH-D-25-01307

Prevalence and risk factors of birth asphyxia at Livingstone University Teaching Hospital

Dear Dr. Masenga,

Thank you for submitting your manuscript to PLOS Global Public Health. After careful consideration, we feel that it has merit but does not fully meet PLOS Global Public Health’s publication criteria as it currently stands. Therefore, we invite you to submit a revised version of the manuscript that addresses the points raised during the review process.

Two reviewers have read your manuscript and provided their comments below. Please revise your manuscript accordingly, and provide a point-by-point response to reviewers upon resubmission.

We look forward to receiving your revised manuscript.

Kind regards,

Sarah Jose, Ph.D.

Staff Editor

Journal Requirements:

1. Please ensure that your Ethics Statement is available in its entirety at the beginning of your Methods section, under a subheading 'Ethics Statement'.

2. We note that there is identifying data in the Supporting Information file ‘BirthAsphyxiaStudy_DATA_2025-04-29_0601x.xlsx’. Due to the inclusion of these potentially identifying data, we have removed this file from your file inventory. Prior to sharing human research participant data, authors should consult with an ethics committee to ensure data are shared in accordance with participant consent and all applicable local laws.

 -Location data

Additional Editor Comments (if provided):

Reviewers' comments:

Reviewer's Responses to Questions

**Comments to the Author**

1. Does this manuscript meet PLOS Global Public Health’s publication criteria?

Reviewer #1: Yes

Reviewer #2: Yes

2. Has the statistical analysis been performed appropriately and rigorously?

Reviewer #1: No

Reviewer #2: Yes

3. Have the authors made all data underlying the findings in their manuscript fully available (please refer to the Data Availability Statement at the start of the manuscript PDF file)?

Reviewer #1: Yes

Reviewer #2: Yes

4. Is the manuscript presented in an intelligible fashion and written in standard English?

Reviewer #1: Yes

Reviewer #2: Yes

Reviewer #1: The manuscript addresses a critical gap in the understanding of birth asphyxia and will be of interest to clinicians, policymakers, and public health professionals. Below are my comments and suggestions.

1- Study design cannot be a retrospective cross-sectional. It can be either secondary data analysis of medical records or retrospective chart review.

2- Data analysis plan is inadequate. The criteria for selection of variables for multivariable regression analysis are not described. The p-value cutoff must also be mentioned.

3- Replace univariable with Univariate.

4- If CIs are reported, the p-value is not necessarily required.

5- There are AORs for all variables, a few are not even significant. Are they still in the model? The author needs to clearly describe the process and rationale for keeping variables in multivariable models.

6- Improve flow by reducing repetitive phrasing and fixing grammatical errors.

Reviewer #2: Prevalence and risk factors of birth asphyxia at Livingstone University Teaching Hospital

Abstract: the method section needs revision for better clarity

Introduction: well written

Methodology: The methodology is comprehensive, clearly outlining the study design, setting, population, data collection, variables, and analysis approach. The use of a standardized tool (REDCap) and adherence to the STROBE checklist enhance the study's rigor. The definitions and statistical methods are appropriate for the research objectives. However, minor grammatical refinements and clarification on exclusion criteria (e.g., rephrasing “Mothers with incomplete records congenital anomalies...”) would improve clarity. Overall, the section effectively supports the study’s credibility and reproducibility.

Result: well written, But there are sample size inconsistency in table 1, justify or correct it.

After adjusting for confounders in multivariable models, these same four variables remained independently significant: eclampsia conferred a 19-fold increase in odds (AOR = 19.3; 95% CI: 3.0–123.0; p = 0.0017), very wide CI? Recheck.

Table 2, is it univariate or bivariate? Recheck.

Discussion: well done

Conclusion: well done

References: well done

**Do you want your identity to be public for this peer review?** For information about this choice, including consent withdrawal, please see our Privacy Policy

Reviewer #1: No

Reviewer #2: No

---

## [Editor Report · Decision Letter 1]

3 Nov 2025

PGPH-D-25-01307R1

Prevalence and risk factors of birth asphyxia at Livingstone University Teaching Hospital

Dear Dr. Masenga,

Thank you for submitting your manuscript to PLOS Global Public Health. After careful consideration, we feel that it has merit but does not fully meet PLOS Global Public Health’s publication criteria as it currently stands. Therefore, we invite you to submit a revised version of the manuscript that addresses the points raised during the review process.

We look forward to receiving your revised manuscript.

Kind regards,

Paolo Ivo Cavoretto, MD PhD

Academic Editor

Journal Requirements:

Additional Editor Comments (if provided):

The manuscript is imroved after revision. However there are a few issues deserving clarification befoe publication.

Define abnormal cardiotocography and define prolonged labour as it was associated with higher rates of fetal distress and fetal compomise in labour.

Differentiate abnotmal CTG and fetal distress. What is the difference?

Revise English labguage please. For instance:

Table1. Avoid using questions in the variable definitions. "Did mother suffer prolapsed umbilical cord" to be replaced with "umbilical cord prolapse",. etc

Discuss the issue that normally prolonges second stage is associated with lower umbilical pH and neonatal acidemia and provide references. There are studies showing Quantification of Posterior Risk Related to Intrapartum FIGO 2015 Criteria for Cardiotocography in the Second Stage of Labor and others assessing Hazard and cumulative incidence of umbilical cord metabolic acidemia at birth in fetuses experiencing the second stage of labor and pathologic intrapartum fetal heart rate requiring expedited delivery. Some of this studies show a correlation with timing in the second stage: the longer the second stage the higher the risk of neonatal acidemia. Compare this with their finding and discuss this difference.
---

## [Editor Report · Decision Letter 2]

23 Nov 2025

Prevalence and risk factors of birth asphyxia at Livingstone University Teaching Hospital

PGPH-D-25-01307R2

Dear Prof. Masenga,

We are pleased to inform you that your manuscript 'Prevalence and risk factors of birth asphyxia at Livingstone University Teaching Hospital' has been provisionally accepted for publication in PLOS Global Public Health.

Best regards,

Paolo Ivo Cavoretto, MD PhD

Academic Editor

The revision was thorough and the article may now be tentatively accepted pending only technical checking by the assistant editors.